# Unveiling the Microbiome’s Role in Hidradenitis Suppurativa: A Comprehensive Review of Pathogenetic Mechanisms

**DOI:** 10.3390/ijms26199542

**Published:** 2025-09-30

**Authors:** Catarina Queirós, Carmen Lisboa, Sofia Magina

**Affiliations:** 1Dermatology and Venereology Department, Unidade Local de Saúde de Gaia/Espinho, 4434-502 Vila Nova de Gaia, Portugal; 2Dermatology and Venereology Department, Unidade Local de Saúde de São João, 4200-319 Porto, Portugal; 3RISE-Health, Pathology Department, Faculdade de Medicina da Universidade do Porto, 4200-319 Porto, Portugal; 4Biomedicine Department, Faculdade de Medicina da Universidade do Porto, 4200-319 Porto, Portugal

**Keywords:** hidradenitis suppurativa, microbiome, pathogenesis

## Abstract

Hidradenitis suppurativa (HS) is a chronic, recurrent, and highly debilitating inflammatory disorder of the pilosebaceous unit. Its pathogenesis is considered multifactorial, involving genetic, environmental, hormonal, lifestyle, and microbiome-related factors. The microbiota, defined as the collection of microorganisms, their genomes, and their interactions within a given environment, colonizes multiple sites of the healthy human body, which include the skin and gut, where it contributes to the maintenance of homeostasis. In HS, both skin and gut microbiota exhibit disruptions in composition and diversity, a state referred to as dysbiosis. Alterations in the expression of antimicrobial peptides in HS further implicate the microbiome in disease pathophysiology. In addition, chronic inflammation, bacterial biofilm formation, and dysbiosis are thought to contribute to the severity and recurrence of HS. Although the precise role of dysbiosis in HS pathogenesis remains unclear, several studies have demonstrated a reduction in cutaneous microbial diversity in HS patients, distinguished by an increased abundance of anaerobic and opportunistic bacteria and a reduction in commensal species. The intestinal microbiome has been even less thoroughly investigated, but available evidence suggests decreased overall diversity and richness, with enrichment of pro-inflammatory and depletion of anti-inflammatory bacterial taxa. This review aims to provide an overview of the current knowledge regarding the role of the microbiome in HS, with the goal of informing the direction of future research, including the potential utility of the microbiome as a biomarker for diagnosis and severity stratification in HS.

## 1. Introduction

Hidradenitis suppurativa (HS) is a chronic, recurrent, progressive, and highly debilitating inflammatory disorder of the pilosebaceous unit. Clinically, it is characterized by the presence of nodules, abscesses, fistulas, and scarring, predominantly affecting intertriginous regions such as the axillae, groin, gluteal, and inframammary areas [1,2]. The reported population prevalence is up to 4%, with disease onset typically occurring in early adulthood. HS affects both sexes, with a female-to-male ratio of approximately 3:1 [3,4]. Smoking and obesity are well-established risk factors, and the disease is frequently associated with comorbidities including spondyloarthropathy, metabolic syndrome, polycystic ovary syndrome, and inflammatory bowel disease (IBD), particularly Crohn’s disease (CD). Psychiatric comorbidities, such as depression, anxiety, and substance misuse, are also common due to the significant psychological burden of the disease [4,5,6].

The human microbiota colonizes multiple body sites, including the skin and gastrointestinal tract, where it contributes to homeostasis. In HS, both cutaneous and intestinal microbial communities are disrupted, showing altered composition and reduced diversity—a state known as dysbiosis. Changes in antimicrobial peptides (AMP) levels in HS further reinforce the contribution of the microbiome to disease pathophysiology [7]. Dysbiosis may contribute to the immune dysregulation observed in HS; however, its role remains unclear. It may act as a trigger for inflammation, represent an adaptive response to inflammatory changes, or serve as a factor amplifying disease progression [8].

The pathogenesis of HS is considered multifactorial, involving genetic predisposition, environmental influences, hormonal factors, lifestyle, and the microbiome [3,6,9,10]. Current models suggest that disease initiation occurs at terminal hair follicles, with hyperkeratosis leading to follicular occlusion and dilation, most often in intertriginous regions. Mechanical stress from friction within skin folds further promotes immune activation [3]. Both innate and adaptive immune cells are involved, releasing pro-inflammatory cytokines such as TNF, IL-1α, and IL-17, which amplify local inflammation [11]. These cytokines also stimulate the production of matrix metalloproteinases (MMP-2, MMP-8, and MMP-9), which degrade the extracellular matrix, weakening the follicular basement membrane [1,12]. Follicular rupture subsequently releases bacteria and cellular debris, fueling chronic inflammation and driving the formation of nodules, abscesses, tunnels, and scarring that typify HS [7,9].

This review aims to synthesize current knowledge on the role of the microbiome in HS, particularly its potential contribution as a cause, consequence, or amplifying factor of inflammation. In doing so, we seek to inform future research directions and highlight the potential of the microbiome as a target for the development of novel therapeutic strategies.

## 2. Microbiome

### 2.1. General Concepts

The microbiome is defined as the collective community of microorganisms (including fungi, bacteria, and viruses), their genomes, and their interactions within a specific environment [13,14,15]. Both the skin and the gut are complex immunological and neuroendocrine organs that harbor distinct microbiomes, maintained through continuous exposure to the external environment [16,17]. Disruption of this balance impairs homeostasis and results in dysbiosis [5,9].

With regard to the skin microbiome, four principal bacterial phyla predominate: *Actinobacteria*, *Bacteroidetes*, *Firmicutes*, and *Proteobacteria* [3,18,19]. The skin microbiome contributes to multiple processes, including cytokine regulation, keratinocyte differentiation, epidermal barrier maintenance, activation of local regulatory T-cell responses, and perpetuation of innate immune responses [9,19,20,21,22]. Its composition is shaped by exogenous factors (e.g., skincare products, clothing, humidity, temperature, ultraviolet exposure) and endogenous factors (e.g., age, anatomical site, skin pH, hygiene practices, hair density, sebaceous gland activity, and local moisture) [9,18,22,23,24]. Host–microbiome interactions are mediated by the recognition of microbial-associated molecular patterns (MAMPs) through pattern recognition receptors (PRRs) [3]. In HS, mechanical friction in intertriginous regions induces local tissue damage, resulting in the release of damage-associated molecular patterns (DAMPs) and penetration of bacterial components into the skin. This activates resident immune cells, which secrete pro-inflammatory cytokines such as IL-1 and TNF. These mediators promote hyperplasia and hyperkeratosis of the follicular infundibulum, leading to follicular occlusion, bacterial overgrowth, and stasis. Consequently, inflammatory mediators—including IL-1, IL-17, caspase-1, S100A8, and S100A9—are further upregulated [3].

The gut microbiome participates in diverse physiological processes, including digestion, nutrient synthesis, and immune system modulation. Six bacterial phyla are commonly detected in the human gut: *Actinobacteria*, *Bacteroides*, *Firmicutes*, *Fusobacteria*, *Proteobacteria*, and *Verrucomicrobia*, with *Bacteroidetes* and *Firmicutes* being the dominant taxa in healthy adults [14]. The intestinal microbiome is influenced by diet, body mass index, sex, antibiotic exposure, aging, and psychological stress, among other factors. Alterations in microbial diversity may increase host susceptibility to disease, compromise mucosal immunological tolerance, and contribute to dermatologic conditions such as acne, atopic dermatitis, and psoriasis [16]. Furthermore, reduced microbial diversity is associated with elevated systemic inflammatory markers—including TNF-α, IL-6, IL-8, and C-reactive protein—and with increased matrix metalloproteinase activity (MMP-2, MMP-8, and MMP-9), which are implicated in HS lesion formation [3,12].

Microbiome analysis is typically conducted using either metataxonomics or metagenomics [15]. Metataxonomics is a rapid and cost-effective method that relies on sequencing marker genes, such as 16S rRNA, which contain hypervariable regions for species identification and conserved regions for primer targeting [24]. The V1–V3 region is recommended for skin microbiome studies, whereas the V4 region is optimal for gut microbiome analysis [5]. However, this method cannot detect viral or fungal sequences and offers limited taxonomic resolution. By contrast, metagenomics sequences all genetic material within a sample without prior locus selection, allowing for de novo genome assembly and high-resolution taxonomic and functional analysis. This approach, however, requires extensive sequencing depth, reference genome alignment, and incurs greater costs, while also potentially capturing host-derived reads [9,20,25,26].

For skin microbiome research, samples can be collected via swabbing, scraping, or tape stripping. Pore strips or cyanoacrylate follicular biopsies are particularly useful for studying follicular communities, while punch biopsies—though most invasive—provide deeper insights [5]. Nielsen et al. demonstrated significant differences in both alpha diversity (species richness within a sample) and beta-diversity (variation between samples) depending on whether swabs or biopsies were used at the same anatomical site in the same individual. This highlights the importance of carefully selecting sampling methods when designing microbiome studies in dermatology [27]. demonstrated significant differences in both alpha diversity (species richness within a sample) and beta-diversity (variation between samples) depending on whether swabs or biopsies were used at the same anatomical site in the same individual. This highlights the importance of carefully selecting sampling methods when designing microbiome studies in dermatology [17,28].

### 2.2. Dysbiosis

Dysbiosis refers to an imbalance in the microbiome and is associated with altered immune system reactivity, thereby predisposing individuals to the development of inflammatory diseases [9,14,15,22]. In HS, the high density of pilosebaceous-apocrine units, combined with increased temperature, elevated moisture, and reduced oxygen availability, creates a microenvironment that favors a distinct microbial composition and greater risk of dysbiosis. Systemic factors such as obesity, diabetes, and nicotine use may further exacerbate this imbalance and contribute to HS pathogenesis [11]. Dysbiosis also appears to influence disease severity: overgrowth of anaerobic bacteria reduces populations of commensal organisms with antimicrobial properties, thereby facilitating the proliferation of pathogenic species [7,29]. Anaerobic bacteria found within skin lesions may promote bacterial expansion and subsequent disruption of follicular units, leading to inflammation. Thus, the altered microbiome in HS contributes to pathogenesis primarily by eliciting an aberrant immune response, rather than by a simple infectious process [18]. Biofilm formation is another important factor in HS. Biofilms are structured microbial communities embedded in an extracellular matrix that confer survival advantages and resistance to conventional antibiotics. They are implicated in the pathogenesis of several dermatologic conditions, including acne and atopic dermatitis [30]. In HS, biofilm formation—particularly by *Staphylococcus epidermidis*—is associated with local accumulation of CD4+ T cells that stimulate regulatory T-cell activity, further perpetuating skin dysbiosis [1].

Microbiome dysbiosis also contributes to HS pathogenesis through alterations in AMPs [7]. AMPs, which may have pro-inflammatory or anti-inflammatory effects, are key components of innate immunity [31]. In healthy skin, AMPs exhibit antimicrobial activity and regulate cytokine production, neutrophil recruitment, antigen presentation, and wound repair. Keratinocytes produce AMPs such as human β-defensin-3 (HBD-3) and psoriasin, while epithelial appendages secrete dermcidin and circulating leukocytes generate human cathelicidin LL-37. In severe HS lesions, HBD-3 levels are suppressed, increasing susceptibility to bacterial superinfection. Psoriasin and dermcidin may also be downregulated, supporting persistent infection and inflammation [11]. Interleukin-22 (IL-22), a cytokine critical for AMP induction, is relatively deficient in HS, and this deficiency may underlie AMP dysregulation [32]. Conversely, LL-37 exhibits predominantly pro-inflammatory activity, including Th1/Th17 cell recruitment and upregulation of inflammatory cytokines. Elevated LL-37 expression has been observed in HS lesions and is also associated with other chronic inflammatory diseases [11]. Thus, LL-37 may be involved from the early stages of HS pathogenesis—initially produced by neutrophils—and may persist into advanced stages, sustaining Th1/Th17-driven inflammation [33].

Gut dysbiosis also negatively influences skin homeostasis through the production of metabolites such as free phenol, p-cresol, and aromatic amino acid derivatives [34]. In addition, intestinal dysbiosis activates systemic inflammatory pathways, leading to elevated cytokine levels (e.g., TNF-α, IL-1β, IL-17) that contribute to HS lesion development by enhancing matrix metalloproteinase expression [12]. Dietary factors, particularly high-fat intake, have been linked to gut dysbiosis, resulting in reduced intestinal production of anti-inflammatory AMPs and increased systemic cytokine release, thereby promoting HS pathogenesis [12]. Specific bacterial taxa may also play a role: for instance, *Bilophila*, an immunogenic genus implicated in IBD, is enriched in HS, whereas *Lachnobacterium*, a butyrate-producing and anti-inflammatory genus, is depleted in the gut microbiome of HS patients [7].

### 2.3. Gut–Skin Axis

The concept of the gut–skin axis was first introduced by Stokes and Pillsbury in 1930, who hypothesized that alterations in the gut microbiome could promote systemic inflammation and thereby exacerbate skin disease [3] (Figure 1). In HS, the potential relevance of this axis was initially suggested by studies showing clinical improvement in patients adhering to a brewer’s yeast–free diet, with disease recurrence upon reintroduction of unrestricted dietary habits [35].

As previously discussed, both the skin and the gut serve as habitats for distinct microbial communities while also hosting large populations of immune cells [16]. Continuous crosstalk between commensal microorganisms and the immune system provides a mechanistic explanation for the frequent coexistence of cutaneous manifestations in nearly one-quarter of patients with primary gastrointestinal disorders [34]. Maintenance of skin homeostasis is closely linked to the integrity of the intestinal barrier, which is reinforced by mucus, immune cells, immunoglobulin A (IgA), and AMPs secreted by epithelial cells. These mechanisms prevent the translocation of gut microbes into the systemic circulation [16].

Several hypotheses have been proposed to explain how gut dysbiosis influences skin pathology. One possibility is that increased intestinal permeability facilitates the migration of bacteria—or bacterial components—from the gut into peripheral tissues. Supporting this, multiple studies have detected bacterial DNA of gut origin in the bloodstream of patients with chronic skin disorders, implicating systemic microbial dissemination in cutaneous inflammation. Another hypothesis suggests that gut microorganisms synthesize neurotransmitters such as norepinephrine, serotonin, and acetylcholine, which stimulate neural pathways and enteroendocrine hormone release. This, in turn, may trigger widespread systemic inflammation with secondary effects on the skin [9]. While compelling, this broader skin–gut–brain axis remains insufficiently understood and requires further investigation.

### 2.4. Cutaneous Microbiome in HS

In the past few years, several studies have focused on the cutaneous microbiome of HS patients. Guet-Revillet and collaborators performed a cross-sectional analysis of the skin microbiome of 82 HS patients with bacterial cultures and metagenomics, identifying two distinct profiles. Profile A was characterized by pure or predominant culture of *S. lugdunensis* and was mostly associated with Hurley stage I lesions. Profile B was represented by a mixed flora composed of Gram-negative and Gram-positive strict anaerobes, anaerobic actinomycetes, and streptococci of the milleri group. This profile was mainly associated with lesions observed in Hurley stages II and III. This was one of the first studies pointing towards different grades of dysbiosis according to disease severity [36].

Ring and collaborators conducted a case–control study, investigating the axillary skin microbiome in 24 HS patients and 24 healthy controls. Peptide nucleic acid (PNA)-FISH probes that target specific bacterial ribosomal RNA sequences in the bacterial cells and confocal laser scanning microscopy were employed. The results showed fewer bacteria and less biofilm in HS skin compared with the control skin. The authors postulated that reduced microbiome in patients with HS may play an important role in the early course of the disease. Moreover, the high level of presence of cocci-biofilm seen among healthy controls highlights the potential biological importance of cocci species, reduced or absent in HS patients [37].

Additionally, Ring. et al. also obtained punch biopsy specimens in order to perform NGS targeting 16S and 18S (V3–V4) rRNA in 30 subjects with HS and 24 healthy controls. In HS patients, biopsy specimens were obtained from non-lesional skin and lesional skin from axilla or groin, whereas in healthy controls specimens were obtained from the axilla only. Regarding lesional skin, identified microbiome types consisted of *Corynebacterium*, *Porphyromonas*, and *Peptoniphilus* spp. On the opposite side, non-lesional skin was dominated by *Acinetobacter* and *Moraxella* spp. Moreover, in healthy controls, *Porphyromonas* and *Peptoniphilus* spp. were not detected, and in HS patients *C. acnes* was not found. The authors found no significant correlation between the number of species and the duration of lesions or its diameter, and no difference in richness between the three groups was observed [38].

Moreover, Ring and collaborators also explored the presence of bacterial biofilm in HS patients by analyzing skin biopsies of 42 individuals in another cross-sectional study. The majority (67%) of lesional samples contained biofilms, which were particularly large in regions such as the infundibulum and sinus tracts, suggesting that biofilm plays a role in chronic lesions in HS, by promoting the chronicity as well as the recalcitrance towards antibiotics [39].

Guet-Revillet et al. decided to analyze samples of pus from draining lesions (collected with swabs) and non-draining lesions (via punch biopsy or needle aspiration) of 65 HS patients, using culture methods and 16S (V1–V2) rRNA sequencing. Comparison with samples from clinically unaffected skin folds from the same participants was performed. The results showed that anaerobes were increased in lesional skin of HS patients, namely with abundance of *Prevotella*, *Porphyromonas* and *Fusobacterium*. Moreover, *Fusobacterium*, *Parvimonas*, *Streptococcus*, and bacteria of the *Clostridiales* order were the most abundant in patients with Hurley stage III disease, thereby confirming that disease severity is associated with marked dysbiosis [40].

Later, Ring et al. used next generation sequencing (NGS) to investigate the bacterial composition of the luminal material found in HS tunnels. This cross-sectional study included 32 HS patients with tunnels in the groin (*n* = 17) or in the axilla (*n* = 15). Swab samples of the luminal material were taken during deroofing procedure, and subjected to NGS targeting 16S (V3–V4) rRNA. *Porphyromonas* spp. and *Prevotella* spp. were the most frequent genera in tunnels, with *Corynebacterium* spp., *Staphylococcus* spp., and *Peptoniphilus* spp. are also frequently present. The association with *Porphyromonas* spp. or *Prevotella* spp. was particularly significant; as these two genera have previously been described in early and suppurating lesions, the authors suggest that they might be associated with the pathogenesis of HS, either as drivers or as biomarkers [29].

In 2020, Riverain-Gillet et al. evaluated the microbiome of clinically unaffected skin in 60 patients with HS and 17 healthy controls. Samples were collected using a sterile swab (rubbing a 5-cm^2^ skin surface for 30 s), and bacterial cultures were subsequently performed. Then, samples were examined by 16S (V1–V2) rRNA gene amplicon sequencing. The authors found that, in terms of bacterial richness, there were no differences between HS skin and the skin of healthy controls. However, they also found that skin of HS patients was characterized by an increased abundance of anaerobes, mostly *Prevotella*, as well as *Actinomyces*, *Mobiluncus*, and *Campylobacter ureolyticus*. On the other side, a lower abundance of skin commensals, such as *Staphylococcus epidermidis* and *Staphylococcus hominis*, was observed. This preclinical dysbiosis also seems to be associated with BMI and with the analyzed anatomic area. The authors therefore suggest that bacterial dysbiosis may be a precocious event in HS, prior to lesion formation [41].

Schneider et al. also evaluated 11 HS patients and 10 healthy controls, by collecting cyanoacrylate follicular biopsy (glue-based) and swab samples from the axilla and groin regions. In HS patients, both non-lesional and lesional skin were sampled. NGS of the 16S rRNA V3–V4 region was performed, and the authors also used a metagenomic prediction tool that allows users to predict metagenome functional content from marker genes. No significant difference in alpha diversity was observed; however, a significant loss of beta-diversity was observed in both non-lesional and lesional skin compared to normal skin. No difference was found in bacterial composition between non-lesional and lesional skin. *Peptoniphilus* and *Porphyromonas* were more abundant in HS skin, and *Cutibacterium* was more abundant in controls (18.8% vs. 1%). Moreover, in HS patients, a trend in altered beta-diversity emerged between smokers and non-smokers, as well as between alcohol users and non-users, suggesting that lifestyle factors may impact skin microbiome. Metabolic pathways were particularly impacted in HS relative to normal skin, indicating that metabolic dysfunction is likely present in HS skin. The genera that contributed to specific metabolic pathways differed between HS and normal skin: *Cutibacterium* contributed significantly to both propanoate metabolism and retinol metabolism in normal skin while *Corynebacterium* was the dominant contributor to propanoate metabolism in HS skin, but played little role in retinol metabolism [42].

In another study, Ring et al. showed significant and systematic differences in the microbiome of various groups (lesional vs. non-lesional; lesional vs. healthy controls), stressing that non-lesional and healthy control samples were only moderately different. Moreover, the authors found 420 differentially abundant genes between lesional samples and healthy controls and 1120 between lesional and non-lesional skin. Between non-lesional skin and healthy controls, only 8 genes were distinguished. PICRUSt data identified cell growth and division as strongly associated with lesional samples. These analyses provided evidence that the key pathways involved in metabolism and genetic information processing are impacted in HS skin microbiome. Overall, increased activity in these pathways may suggest increased microbial proliferation and turnover [43].

Additionally, Naik and collaborators also investigated the correlation between HS severity and dysbiosis, by 16S (V1–V3) rRNA sequencing samples of 12 HS patients and 12 healthy controls taken from the axilla, gluteal crease, inguinal crease, and submammary fold. An increased abundance of Gram-positive and negative anaerobes (such as *Porphyromonadeacea*, *Prevotellaceae*, *Fusobacteria*, and *Clostridales*) and a decreased abundance of *Cutibacterium* spp. was found in HS specimens, prompting the authors to conclude that skin bacterial communities in HS are different from those of controls, with a greater disease severity associated with increased skin bacterial perturbations in HS patients. Indeed, a correlation between increased disease severity, increased relative abundances of anaerobes, and decreased relative abundances of major skin commensals was found [44].

In 2022, McCarthy et al. studied skin and nasal swabs from 59 HS patients and 20 healthy controls. In general, the overall microbiome composition of nasal and skin samples was typical of what has been previously described, mainly composed of the genera *Staphylococcus* and *Corynebacterium*. In HS patients, both nasal and skins swabs showed a reduction in alpha diversity, suggesting a reduction in the richness of skin microbiome compared with those of controls [4]. This reduction in alpha diversity has also been shown in other skin conditions such as atopic dermatitis and psoriasis [45]. Regarding beta-diversity, there was a statistically significant separation with respect to the axilla, groin, and nasal microbiome, showing that different microbiome communities are present at these body sites [4]. *Peptoniphilus lacrimalis*, *F. magna*, *P. coxii*, *Anaerococcus murdochii*, and *A. obesiensis* were more abundant in individuals with HS, with a higher abundance of *Cutibacterium acnes* in healthy controls. This depletion of *C. acnes* suggests that it does not play a pathogenic role in HS as it does in acne. It may, however, alter the microbial ecology of the skin and promote HS pathogenesis indirectly [4]. In particular, *Finegoldia magna* has been shown to have immune-modulating activities, as it can promote the formation of neutrophil extracellular traps, whose abundance is correlated with HS severity [46]. Moreover, *F. magna* has been shown to activate mast cells and basophils, leading to the production of pro-inflammatory cytokines, and to activate pro-inflammatory neutrophils via virulence factors protein L and FAF (*F. magna* adhesion factor) [47].

In order to investigate the impact of treatment on skin microbiome, Hsu et al. examined 22 Asian HS patients and 12 controls before and after adalimumab treatment. *Prevotella* spp. and *Peptoniphilus* spp. were dominant in HS lesional skin, whereas *Paucibacter* spp. and *Caulobacter* spp. were significantly more abundant in health controls. After adalimumab treatment, no significant taxa differences were identified. The authors postulate that dysbiosis in severe HS patients may result from permanent skin damage, which does not recover even after effective treatment. Nonetheless, the small sample size may have influenced the results [48]. Regarding surgical treatment, Pardo and collaborators analyzed samples from 32 HS patients collected during routine surgery, and found that the deeper layers of the skin, captured by biopsies, carry specific microbiome niches colonized by a limited subset of bacteria compared with the external epidermis. In line with previous studies, *Prevotella* and *Porphyromonas* were the most prominent genera in HS lesional skin, with the highest abundance in Hurley III patients. The relative abundance of *Corynebacterium* and *Staphylococcus*, which were the main components of skin swabs, decreased significantly in deep biopsies. In contrast, *Porphyromonas* and *Prevotella* increased in deep biopsies relative to their levels in swabs. The authors concluded on the existence of a marked shift at both phylum and genus levels between swabs and deep biopsies, further diversified by different Hurley stages [49].

In addition, Zhu et al. performed a comprehensive two-sample Mendelian randomization analysis to determine the causal effect of skin and gut microbiome on skin appendage disorders. Regarding skin microbiome in HS, the only association found was with *Staphylococcus hominis*, that seems to be associated with a protective effect on the risk of HS. The authors conclude that the specific mechanisms and modes of intervention for these causalities need to be further explored [50].

Similarly, Guo et al. performed a comprehensive two-sample Mendelian randomization analysis to investigate the causal relationship between skin microbiome and HS. A total of 597 participants from two cohorts were included. The order *Burkholderiales* and genus *Enhydrobacter* were identified as protective factors against HS. Future studies are needed to confirm these findings and to understand how this can translate into clinical practice [51].

Finally, and considering fungal microbiome, Ring et al. also analyzed 30 HS patients and 24 controls, via amplicon sequencing of the highly conserved eukaryotic 18S ribossomal RNA gene. The three most frequent species were *Malassezia restricta*, *Saccharomyces cerevisiae* and *M. globosa*, with no significant differences between HS samples and control samples [52].

Research that used traditional culture methods also found differences in the bacterial composition of HS patients compared to healthy controls. The major findings included the identification of *Staphylococcus aureus*, *Diphtheroid*, *Escherichia coli*, and coagulase-negative staphylococci as the most common bacteria in HS patients, as well as the existence of a correlation between higher Hurley stages and more polymicrobial flora in culture-based studies [7]. However, it is important to note that culture methods have several limitations, the largest one being its low sensitivity [2].

In sum, and despite the fact that these studies used different sampling and sequencing techniques, making it difficult to pool their results, a decrease in microbiome diversity in HS patients seems to be present in almost all studies, when compared to healthy controls [9]. Moreover, an increased abundance of anaerobic bacteria and opportunistic pathogens, including *Porphyromonas*, *Peptoniphilus*, *Bacteroides*, *Peptostreptococcus*, *Prevotella* and *Pseudomonas* spp. and a loss of skin commensal species, such as *Cutibacterium*, in HS lesions has been a consistent finding [5,21,53]. Two major bacterial skin species, *Staphylococcus aureus* and *Streptococcus pyogenes*, which are ubiquitous in normal skin, do not seem to contribute to the pathogenesis of HS [3,26]. *Cutibacterium* is usually decreased in HS lesions; in normal conditions, it produces propionic acid which lowers the pH of the surrounding microenvironment and has antimicrobial activity against other bacteria, thereby potentially contributing to dysbiosis in this condition [26]. Increased prevalence of *Prevotella* and *Porphyromonas* spp. in the skin of HS patients may be involved in the pathogenesis of the disease via upregulation of AMP secretion. This leads to an increase in keratinocyte proliferation and recruitment of neutrophils and macrophages, in turn culminating in follicular occlusion and an increase in TNF-α and NF-kB levels [3,31]. Moreover, increased abundance of *Prevotella* spp. triggers T helper 17 (Th17) immune responses and activates Toll-like receptor 2 (TLR-2), leading to an increased production of IL-1 and IL-23, both involved in the pathogenesis of HS [54]. In addition, chronic HS lesions were shown to contain bacterial biofilms as compared with skin of healthy controls from corresponding areas, and less biofilm was observed in clinically uninvolved skin in matched areas [39].

It is also clear that the microbiome of HS lesions significantly differs from the microbiome found in the same patients’ non-lesional skin folds, as well as from the microbiome of controls [7]. Indeed, non-lesional skin from HS patients also demonstrates decreased abundance of major skin commensals and an increased abundance of anaerobes, reflecting a milder state dysbiosis compared with lesional skin [41]. To date, no definite relationship has been found between the number of species and the duration or size of HS lesions; however, the presence of certain microbes seems to be correlated with the severity of HS manifestations [38] (Table 1 and Figure 2). The main limitation of these studies is the absence of an associated intervention, particularly a therapeutic one, that could influence the microbiome of these individuals.

### 2.5. Intestinal Microbiome in HS

Besides skin microbiome, fecal microbiome has also been the focus of several studies in patients with HS. Eppinga et al. performed a cross-sectional study and evaluated *Faecalibacterium prausnitzii* and *E. coli* levels in patients with psoriasis (*n* = 29), concomitant psoriasis and IBD (*n* = 13), HS only (*n* = 17), and concomitant HS and IBD (*n* = 17). *F. prausnitzii* is important for maintaining gut homeostasis, playing a role in reducing oxidative stress and demonstrating anti-inflammatory properties. On the other side, *E. coli* is capable of adhering to epithelial cells, invade and replicate within macrophages, without triggering host cell death, leading to the release of large amounts of tumor necrosis factor-α. In this study, increased levels of *E. coli* and decreased levels of *F. prausnitzii* were noted in patients with psoriasis, concomitant psoriasis and IBD, and concomitant HS and IBS, but not in patients with HS only [55].

Later, Kam et al. evaluated fecal samples of three patients with HS Hurley stage II or III and 3 controls. The samples underwent bacterial 16S rRNA sequencing, and results revealed that HS seems to be associated with decreased gut microbiome species diversity, increased abundance of *Bilophila* and *Holdemania*, and reduced abundance of protective *Lachnobacterium* and *Veillonella*. Moreover, the phylum *Firmicutes* was noted to be reduced in HS patients, as compared to controls. Nevertheless, the authors underlined the fact that smoking can reduce the relative abundance of *Firmicutes* in the intestine, and many HS patients are smokers [56].

In 2021, Lam et al. performed 16S rRNA sequencing in fecal samples from 17 HS patients and 20 healthy participants, finding no differences between HS patients and controls in bacterial richness, Shannon, and inverse Simpson indices, nor in bacterial community structure based on Bray–Curtis or Jaccard metrics. Moreover, no significant differences in alpha-and beta-diversity were detected when stratified by BMI or smoking status. However, considerable taxonomic differences between feces of HS patients and healthy controls were found. The most striking finding was the presence of *Robinsoniella* in feces of the majority of HS patients and none of the healthy controls. *Sellimonas* was also more common in patients than controls. An IBD-like microbial signature with a decrease in *Faecalibacterium prausnitzii* was not found, and a member of *Christensenellaceae* family was depleted in fecal samples of HS patients compared to healthy controls. *Mesorhizobium* was present in all HS skin samples and, in particular, in lesional compared to non-lesional skin, but the relevance and robustness of this genus remains uncertain [57].

McCarthy et al. also evaluated fecal samples of 59 HS patients, comparing the results with fecal samples of a control group consisting of 30 participants. The samples underwent bacterial 16S rRNA gene amplicon sequencing on total DNA. The authors found that alpha diversity was significantly lower in patients with HS. This decrease was also observed for other metrics of alpha diversity, including Shannon and phylogenetic diversity. Comparison of global microbial composition in all the samples (beta-diversity) revealed a microbiome separation between HS subjects and healthy controls in all metrics tested, noting also less clustering within the HS samples. The authors also found that one of the greatest differences between patients with HS and healthy controls was elevated levels of *Ruminococcus gnavus* and *Clostridium ramosum* in the first group [4]. Moreover, *R. gnavus* seems to be overrepresented not only in HS participants, but also in patients with CD, and is associated with spondylarthritis and irritable bowel syndrome [58,59,60]. In particular, its role in CD has been supported by the production of a pro-inflammatory polysaccharide, which increases the production of TNF-α by interacting with toll-like receptor 4 (TLR4) of immune cells [61]. Therefore, the authors postulated that the production of this polysaccharide may as well contribute to the pathogenesis of HS. Thus, it is possible that the comorbidities associated with HS have a common etiology, indirectly due to the activity of *R. gnavus*. Notably, *C. ramosum* has also been reported to be increased in CD and obesity [62]. The authors note that the use of antibiotics in HS may play a role in the reduction in alpha diversity seen in their study; however, they did not detect any significant difference in amplicon sequence variants between those who received antibiotics in the preceding year and those who did not [4].

Additionally, Ogut and collaborators evaluated fecal microbiome of 15 HS patients and 15 controls, and found that alpha diversity of the gut microbiome was significantly lower in HS patients compared to healthy individuals. Although gut microbiome of both HS patients and healthy controls was largely dominated by *Firmicutes*, *Bacteroidetes*, *Actinobacteria*, *Proteobacteria*, *Verrucomicrobia* and unclassified bacteria, beta diversity was significantly different between patients with HS and healthy controls. *Firmicutes* was the most predominant phylum among all relatively abundant dominant taxa in HS and healthy controls. HS patients showed a significantly reduced abundance of three genera: unclassified *Clostridiales*, *Fusicatenibacter*, and unclassified *Firmicutes* (all belonging to *Firmicutes* phylum). This low abundance may be a triggering factor for systemic inflammation through decreased SCFA production and dysregulation of inflammatory mechanisms, shifting toward a pro-inflammatory state. No significant differences were found in gut microbiome between non-obese and obese, non-smokers and smokers, and treatment naïve and under treatment groups, but the small sample size may have influenced these results [63].

In 2023, Cronin et al. analyzed fecal microbiome of 55 HS patients, 102 CD patients and 95 controls. Although clear differences existed between the fecal microbiome of HS and CD patients, both diseases can be characterised by a shared set of 11 genera which are significantly enriched compared to the control group. These include *Streptococcus*, *Veillonella*, *Eggerthella* and *Anaerotruncus*, amongst others. They also shown that the microbiome of HS and CD is characterised by a shared set of depleted taxa compared to controls. These include multiple genera in the *Ruminococcaceae* and *Lachnospiraceae* families as well as genera such as *Coprococcus* and *Christensenellaceae* group. Moreover, these authors identified two groups of HS patients: one with a gut microbiome similar to CD (enriched with genera such as *Escherichia*_*Shigella*, *Veillonella*, *Enterococcus*, *Bacteroides* and *Streptococcus*, known to promote intestinal inflammation, and with lower levels of *Faecalibacterium* spp., known to be anti-inflammatory), who may be in a higher risk of developing IBD, and another one with a gut microbiome more similar to normal controls, with higher abundance of the *Ruminococcus* cluster and the *Lachnospiraceae* cluster. These findings highlight the potential of use of the fecal microbiome as a biomarker in identifying patients with HS at higher risk of developing CD. Additionally, these authors also detected several associations with inflammatory markers in those patients with HS and a gut microbiome profile similar to CD. These include elevated levels of IL-12, a cytokine known to be implicated in CD pathogenesis, and lower levels of the Gas6 protein, known for its anti-inflammatory activity through activation of TAM receptors present on activated T regulatory cells [64].

More recently, Liu et al. performed a bidirectional two-sample Mendelian randomization analysis using genome-wide association study summary data of gut microbiome and HS from the MiBioGen Consortium. In this study, family XI (a family in *Clostridiales* also known as *Clostridium* cluster XI) and *Porphyromonadaceae* were identified as preventive in HS, whereas *Clostridium innocuum* group and *Lachnospira* were associated with a higher risk of HS. Those preventive phyla include *Bacteroidetes* and *Clostridiales*, which are capable of producing short-chain fatty acids (SCFAs) and thought to preserve intestinal barrier function by preventing the passage of pro-inflammatory molecules into the systemic circulation, thereby preventing metabolic endotoxemia. *Lachnospira* group, on the other way, might worsen HS by triggering systemic inflammation in the face of a dysbiotic environment [65].

Lelonek et al. also analyzed fecal microbiome in 80 HS patients and 80 controls. No differences in alfa diversity were found between the groups, but the authors identified significant disparities in dietary habits between individuals with HS and healthy controls, with the former exhibiting elevated sugar consumption and a higher prevalence of milk consumption [66].

In the previously mentioned Mendelian randomization analysis of Zhu et al., and now regarding gut microbiome, phylum *Lentisphaerae* and family *Prevotellaceae* were associated with an increased HS risk. Conversely, genus *Bifidobacterium*, *Eubacteriumfissicatenagroup*, and *Fusicatenibacter* were associated with a protective effect on HS risk. Again, the authors state that these causalities need to be further explored [50].

Finally, patients with HS have been shown to have lower vitamin D levels, and these seem to be correlated with disease severity [67]. Vitamin D controls proliferation and differentiation of the epidermis and its structures (including hair follicles), and is able to regulate skin homeostasis and immune responses. As vitamin D metabolism is, at least, partly regulated by intestinal microbiome [68], one can speculate that low vitamin D levels noted in HS patients could be a consequence of their intestinal dysbiosis [3].

In sum, and however more research is needed in the field of gut microbiome and HS, some common trends can be identified, such as a lower overall diversity and richness of the gut microbiome, a higher abundance of pro-inflammatory bacteria, such as *Proteobacteria* and *Actinobacteria*, and a lower abundance of anti-inflammatory bacteria, such as *Firmicutes* and *Bacteroidetes*, in HS patients compared to healthy controls. Moreover, the presence of certain bacterial species (such as *Robinsoniella peoriensis*, *Bilophila*, *Holdemania*, and *Ruminococcus callidus*), suggests potential roles in the pathogenesis of HS [9] (Table 2 and Figure 2). Again, one of the major limitations of these studies is the absence of an associated intervention that could influence gut microbiome of these individuals.

### 2.6. Blood Microbiome in HS

Regarding blood microbiome, the results of the few studies performed have proved contradictory. Ring et al. analyzed blood microbiome in 27 HS patients and 26 healthy controls using two methods: NGS of 16S ribosomal RNA, and routine anaerobic and aerobic blood culturing. Blood culturing samples were negative. Due to its higher sensitivity, several bacteria including *AcinetobacSter* and *Moraxella* were detected using NGS. However, HS patients’ peripheral blood did not differ significatively in bacterial composition from that of healthy controls [69].

Later, Patricia Hispán et al. assessed the presence of bacterial DNA in the peripheral blood of 50 HS patients and 50 healthy controls, finding that bacterial DNA was significantly more likely to be detected in HS patients (34% vs. 2% in controls). The most frequently detected were Gram-negative bacteria, especially *Escherichia coli*, which is a member of the intestinal microbiome of healthy individuals. Moreover, the presence of bacterial DNA in the serum was associated with elevated levels of pro-inflammatory cytokines, mainly TNF-α and IL-17A. Therefore, the authors hypothesized that pathogenesis of HS may be affected by bacterial translocation from the intestinal lumen, as proposed in other inflammatory conditions [70].

Therefore, and although it may be hypothesized that blood cytokine response may be influenced by the presence of bacterial DNA in serum, more studies are needed to explore this association [3].

### 2.7. Methodological Limitations

Despite the growing body of research, most available studies have included relatively small cohorts, limiting the strength of statistical analyses and the generalizability of findings. Furthermore, heterogeneity in methodological approaches—including NGS, 16S rRNA sequencing, and conventional culture techniques—further complicates cross-study comparisons. These limitations highlight the need for standardized methodologies and larger, well-designed cohorts in future investigations.

## 3. Comorbidities in HS and Microbiome

### 3.1. Obesity and Smoking

Obesity and smoking are among the most frequent comorbidities in patients with HS [4,10]. Obese HS patients often exhibit pronounced dysbiosis, characterized by alterations in microbial diversity, composition, and functionality. Such imbalances may contribute to a chronic pro-inflammatory state that favors HS development. Specifically, an increased abundance of *Staphylococcus* species has been reported in the skin microbiome of obese individuals. In the gut, these patients frequently demonstrate dysregulation of *Prevotella* and *Bacteroides* species, along with an elevated Firmicutes-to-Bacteroidetes ratio. In contrast, non-obese HS patients generally display less severe microbiome alterations [9,12]. Beyond microbial changes, obesity also increases the extent of skinfolds and mechanical stress, creating anaerobic microenvironments that promote bacterial growth and further stimulate inflammation [7].

Smoking has similarly detrimental effects on the microbiome and cutaneous immunity. Nicotine promotes follicular acanthosis, facilitating follicular plug formation, while also suppressing antimicrobial peptide production, thereby increasing susceptibility to infection [7,10]. In addition, smoking supports the proliferation of *Staphylococcus aureus*, which disrupts the cutaneous microbiome and drives local inflammatory responses [1]. At the gastrointestinal level, smokers tend to exhibit increased relative abundance of *Proteobacteria* and *Bacteroidetes*, coupled with reductions in *Actinobacteria* and *Firmicutes*. Smoking has also been shown to decrease overall gut microbial diversity, potentially through mechanisms such as altered acid–base balance, increased oxidative stress, and disruption of intestinal tight junctions and mucin composition [3].

### 3.2. Inflammatory Bowel Disease

HS and IBD share several clinical and immunological features. Both conditions are characterized by sterile abscess formation, sinus tract development, and overlapping inflammatory pathways, including dysregulated Th1 responses and increased production of cytokines such as IL-1, IL-6, IL-17, IL-23, and TNF-α [4,53,64,71,72]. Smoking and obesity are common risk factors for both diseases, and clinical improvement has been observed with TNF-α inhibitors in each [3,4]. Anti-*Saccharomyces cerevisiae* antigen (ASCA) IgGs are associated with IBD, and these antibodies have been identified in patients with HS, particularly those with severe disease, further supporting a possible association between both conditions. ASCA antibodies, a serological marker of IBD, have also been detected in HS patients—particularly those with severe disease—further supporting a pathogenic link between the two disorders [2]. Consistent with this, the prevalence of IBD among HS patients is estimated at 3.3%, representing a 4- to 8-fold increase compared with the general population [73]. Conversely, the prevalence of HS has been reported at 26% in patients with CD and 18% in those with ulcerative colitis [74].

Alterations of the gut microbiome are well documented in CD and are increasingly being investigated in HS. As previously noted, Cronin et al. identified two subgroups of HS patients: one with a microbiome profile comparable to healthy controls, and another with a profile closely resembling that of CD patients. These findings suggest that a subset of HS patients may harbor gut microbiome configurations similar to those seen in CD, potentially conferring an elevated risk for developing CD [64].

Additional metabolic pathways may also contribute to the overlap between HS and IBD. Homocysteine (Hcy), a metabolite directly implicated in inflammatory processes, is elevated in multiple inflammatory dermatoses, including HS, acne vulgaris, and psoriasis [12]. Marasca et al. demonstrated that Hcy levels are significantly higher in HS patients and positively correlate with disease severity, as measured by the Sartorius score [75]. Furthermore, pathways involved in D-glucarate and D-galactarate degradation appear to be overrepresented in the fecal microbiome of HS patients. These pathways are likewise enriched in CD, where they are associated with systemic inflammation and poor prognosis [4].

At present, it remains unclear whether inflammation in HS originates primarily within the skin or whether extracutaneous inflammation precedes cutaneous manifestations. Should the latter be the case, alterations in the intestinal microbiome may play a pivotal role in disease initiation, offering a plausible explanation for the observed association between HS and CD [64].

## 4. Future Directions

Biomarkers are defined as parameters that can be objectively measured and evaluated to indicate normal biological processes, pathogenic processes, or pharmacological responses to therapeutic interventions [17]. Given the potential association between HS and bacterial dysbiosis, the identification of distinct microbial signatures raises the possibility of using the microbiome as a biomarker for this condition. Such an approach could facilitate both diagnosis and patient stratification. The observed reduction in microbial diversity, enrichment of pro-inflammatory bacterial populations, and correlations with comorbidities underscore the relevance of this line of investigation. Furthermore, longitudinal assessment of the microbiome—both across the natural course of disease and in response to treatment—may provide valuable insights into disease progression and therapeutic outcomes [4,6,9,15,25,28,64]. While important groundwork has been established, substantial advances remain necessary. Future research should prioritize well-designed prospective and randomized studies to validate the microbiome as a clinically meaningful biomarker in HS.

## Figures and Tables

**Figure 1 ijms-26-09542-f001:**
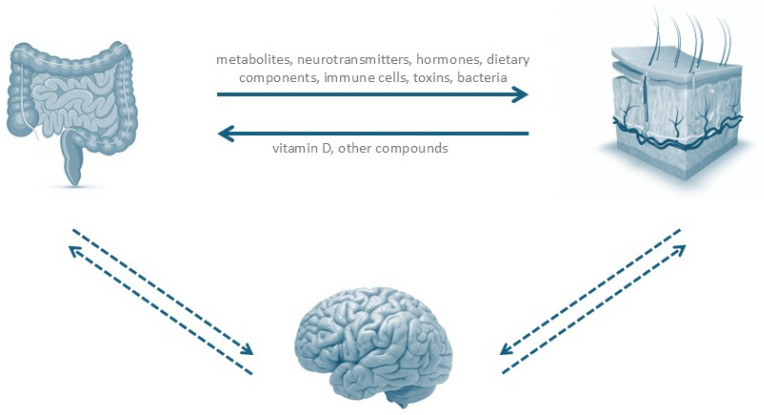
Dysbiosis in the gut can result from dietary components, illnesses, lifestyles, prebiotics, antibiotics, probiotics, or drugs. This state of dysbiosis further impacts intestinal barrier (enabling the passage of microbes to the circulation), promotes the production of toxic compounds and neurotransmitters by gut microbes or by the host, and impacts B and T-cell immunity. The skin also takes part in vitamin D metabolism and produces other chemicals that may influence gut homeostasis. Additionally, emotional states also impact systemic inflammation, explaining the frequent association between skin diseases and states of anxiety/depression and other mood disorders.

**Figure 2 ijms-26-09542-f002:**
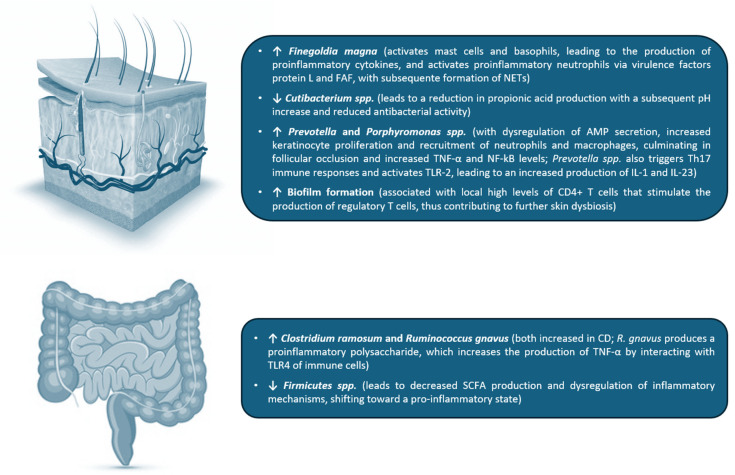
Examples of pathophysiological mechanisms by which cutaneous and intestinal dysbiosis may contribute to the development of HS. FAF: *Finegoldia* adhesion factor; NETs: neutrophil extracellular traps; AMP: antimicrobial peptides; TLR: Toll-like receptor; CD: Crohn disease; SCFA: short chain fatty acids.

**Table 1 ijms-26-09542-t001:** Cutaneous microbiome in HS patients and in normal skin—summary of common differences.

HS PATIENTS	NORMAL SKIN
Decrease microbiome diversity	Increased microbiome diversity
Increased abundance of anaerobic bacteria and opportunistic pathogens, (such as *Porphyromonas*, *Peptoniphilus*, *Bacteroides*, *Peptostreptococcus*, *Pseudomonas* and *Prevotella* spp.)	Decreased abundance of anaerobic bacteria and opportunistic pathogens, ubiquitous presence of *Staphylococcus aureus* and *Streptococcus pyogenes*
Loss of skin commensal species, such as *Cutibacterium*	Presence of skin commensal species, such as *Cutibacterium*
Bacterial biofilms commonly present	Bacterial biofilms usually absent

**Table 2 ijms-26-09542-t002:** Intestinal microbiome in HS patients and in normal gut—summary of common differences.

HS PATIENTS	NORMAL GUT
Lower overall microbial diversity and richness	Increased microbial diversity and richness
Higher abundance of pro-inflammatory bacteria, such as *Proteobacteria* and *Actinobacteria*	Lower abundance of pro-inflammatory bacteria, such as *Proteobacteria* and *Actinobacteria*
Lower abundance of anti-inflammatory bacteria, such as *Firmicutes* and *Bacteroidetes*	Higher abundance of anti-inflammatory bacteria, such as *Firmicutes* and *Bacteroidetes*
Presence of certain bacterial species potential implied in disease pathogenesis (such as *Robinsoniella peoriensis*, *Bilophila*, *Holdemania*, and *Ruminococcus callidus*)	Absence of these species

## Data Availability

Data sharing not applicable to this article as no datasets were generated or analysed during the current study.

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
