# Peer review of "Unveiling the Microbiome’s Role in Hidradenitis Suppurativa: A Comprehensive Review of Pathogenetic Mechanisms"

_ijms, 2025, doi:10.3390/ijms26199542_

Round 1
Reviewer 1 Report
Comments and Suggestions for Authors
Recommendations for review
Abstract and title: The abstract should reflect the pathogenic mechanisms promised in the title, not just descriptive observations. The role of the skin and gut microbiome in: chronic inflammation (IL-1, TNF-α, IL-17); bacterial biofilm formation; dysbiosis contributing to the severity and recurrence of HS should be clearly included. Differences between the skin and gut microbiome and their impact on pathogenesis should be mentioned.
In the Introduction, I would suggest that the discussion of the microbiome be placed immediately after the brief description of HS and general inflammatory mechanisms. The hypotheses of the study should be more clearly highlighted, namely: the microbiome as a cause, a consequence or as an amplifying factor of inflammation. Concrete examples of relevant studies should also be included to justify the purpose of the study, and therefore of the article.
Regarding the Microbiome Analysis, this should include in the skin, gut and blood subsections: clarification of the type of study (observational, MR, cross-sectional); sample size and impact on the interpretation of the results and the limitations of each study (methodological heterogeneity, biases). Direct comparisons between studies should also be added, when possible, with consistent contradictions and trends highlighted.
In the Methodological Limitations and Critique, it would be good to highlight: the small number of participants in many of the analyzed studies and the impact of generalizability; methodological differences (NGS, 16SrRNA, traditional cultures) that make it difficult to compare data; lack of clear evidence on the role of the microbiome. A clear section for study limitations should be added, just before the conclusions.
In the Conclusions, do not assert the causality of the microbiome without presenting solid evidence. The general trends should be presented in summary: decrease in microbiome diversity, increase in pro-inflammatory bacterial populations and association with comorbidities. The need for future prospective and randomized studies should be mentioned, in order to validate the observations.
In the Tables and Figures, better clarity should be ensured, each table should indicate the type of study, the number of participants and the main results. Possibly, clearly separate the skin microbiome from the intestinal microbiome to avoid confusion.
Reviewer 2 Report
Comments and Suggestions for Authors
In this review, the authors attempt to systematize and summarize current evidence on the role of the microbiome in the pathogenesis of hidradenitis suppurativa (HS)—a chronic inflammatory skin disease characterized by recurrent flares and a profound reduction in patients’ quality of life. The topic is highly relevant: over the past decade, interest in the study of skin and gut microbiota has grown substantially, and the inclusion of HS within this research domain appears both timely and promising. The authors highlight the importance of both the cutaneous and intestinal microbiome, emphasizing their interconnectedness and potential contribution to systemic inflammatory responses. Nevertheless, in the context of HS, the skin microbiome represents the key element of pathogenesis, and a more detailed analysis of its alterations—such as the reduction of commensal species, the overgrowth of anaerobic and opportunistic bacteria, and imbalances between specific taxa—would have further strengthened the review. It would also be valuable to elaborate on how these microbial shifts interact with innate immune components of the skin.
For greater clarity, the inclusion of a schematic illustration depicting the interaction between the skin microbiome, the cutaneous barrier, immune cells (keratinocytes, dendritic cells, T cells), and innate immunity factors (e.g., antimicrobial peptides) would have provided readers with a clearer understanding of how dysbiosis may initiate and sustain chronic inflammation.
A further limitation is that the review does not sufficiently address the rapid adoption of long-read sequencing technologies (Oxford Nanopore Technologies, PacBio) that enable full-length 16S rRNA gene sequencing. These approaches provide improved taxonomic resolution down to the species level, which is particularly relevant for the skin microbiome, where differences between closely related taxa may be critical to disease pathogenesis.
The article also presents an illustration (Figure 1) depicting a bidirectional connection between the skin microbiome and the brain. While the concept of the “gut–brain axis” is well established and supported by a substantial body of literature, evidence for a “skin–brain axis” remains limited. More robust empirical data are needed to substantiate this model. The authors should either provide stronger supporting evidence or explicitly acknowledge the existing gaps in this area.
Finally, while the review enumerates observed alterations in microbial composition, it does not sufficiently discuss the mechanistic pathways through which these changes are translated into inflammatory cascades that disrupt immune homeostasis in the skin. Moreover, a discussion of potential therapeutic strategies aimed at modulating the skin microbiome—such as topical probiotics, bacteriophage therapy, or postbiotic interventions—would enhance the translational value of the work.
Overall, the article addresses a clinically significant and scientifically timely topic, and it provides a useful synthesis of current knowledge. However, its impact could be strengthened by a sharper focus on the skin microbiome, the integration of novel methodological advances such as long-read sequencing, and a more critical discussion of proposed mechanistic models and therapeutic perspectives.
Round 2
Reviewer 1 Report
Comments and Suggestions for Authors
Dear authors,
I congratulate you for your work in revising the manuscript. The revised version (V2) has integrated all major comments: the abstract now reflects the pathogenetic mechanisms, the introduction is more clearly structured and the study hypotheses are explicitly highlighted. The microbiome analysis mentions the study type, sample size and methodological limitations, with direct comparisons between studies and an emphasis on contradictions and trends. You have also added a well-structured “Limitations” section and reworded the conclusions to avoid causality claims, presenting only general trends and the need for prospective/randomized studies.
Tables and figures have been clarified, now contain the study type, number of participants and main results, and cutaneous and intestinal data are separated to avoid confusion. All these changes have led to a substantial improvement in the quality and clarity of the article.
Reviewer 2 Report
Comments and Suggestions for Authors
The Authors are highly appreciated for providing accurate and comprehensive responses. The manuscript has been significantly improved and can be approved for publication in present form.